# FAST ADAPTATION VIA HUMAN DIAGNOSIS OF TASK DISTRIBUTION SHIFT

## ABSTRACT

When agents fail in the world, it is important to understand *why* they failed. These errors could be due to underlying distribution shifts in the goals desired by the end user or to the environment layouts that impact the policy's actions. In the case of multi-task policies conditioned on goals, this problem manifests in difficulty in disambiguating between goal and policy failures: is the agent failing because it can't correctly infer *what* the desired goal is or because it doesn't know *how* to take actions toward achieving the goal? We hypothesize that successfully disentangling these two failures modes holds important implications for selecting a finetuning strategy. In this paper, we explore the feasibility of leveraging human feedback to diagnose *what* vs. *how* failures for efficient adaptation. We develop an end-to-end policy training framework that uses attention to produce a human-interpretable representation, a visual masked state, to communicate the agent's intermediate task representation. In experiments with human users in both discrete and continuous control domains, we show that our visual attention mask policy can aid participants in successfully inferring the agent's failure mode significantly better than actions alone. Leveraging this feedback, we show subsequent empirical performance gains during finetuning and discuss implications of using humans to diagnose parameter-level failures of distribution shift.

## 1 INTRODUCTION

Humans are remarkably adept at asking for information relevant to learning a task (Ho & Griffiths, 2022). This is in large part due to their ability to communicate feature-level failures of their internal state via communicative acts to a teacher (e.g. expressing confusion, attention, understanding, etc.) (Argyle et al., 1973). Such failures can range from not understanding *what* the task is, e.g. being asked to go to Walgreens when they don't know what Walgreens is, to not knowing *how* to accomplish the task, e.g. being asked to go to Walgreens and not knowing which direction to walk in. In both cases, a human learner would clarify *why* they are unable to complete the task so that they can solicit feedback that is most useful for their downstream learning. This synergistic and tightly coupled interaction loop enables a teacher to better estimate the learner's knowledge base to give feedback that is best tailored to filling their knowledge gap (Rafferty et al., 2016).

Our sequential decision-making agents face the same challenge when trying to adapt to new scenarios. When agents fall in the world due to distribution shifts between their training and test environments (Levine et al., 2020), it would be helpful to understand why they fail so that we can provide the right data to adapt the policy. The difficulty today when dealing with systems trained end-to-end is that they are inherently incapable of expressing the cause of failure and exhibit behaviours that may be arbitrarily bad, rendering a human user left in the dark with respect to what type of feedback would be most useful for finetuning. Ergo, active learning strategies focus on generating state or action queries that would be maximally informative for the human to label (Akrour et al., 2012; Bobu et al., 2022; Reddy et al., 2020; Bıyık et al., 2019), but such methods require an unscalable amount of human supervision to cover a large task distribution (MacGlashan et al., 2017).

To address the challenge above, we propose a human-in-the-loop framework for training an agent end-to-end capable of explicitly communicating information useful for a human to infer the underlying cause of failure and provide targeted feedback for finetuning. In the *training phase*, we leverage attention to train a policy capable of producing an intermediate task representation, a masked state

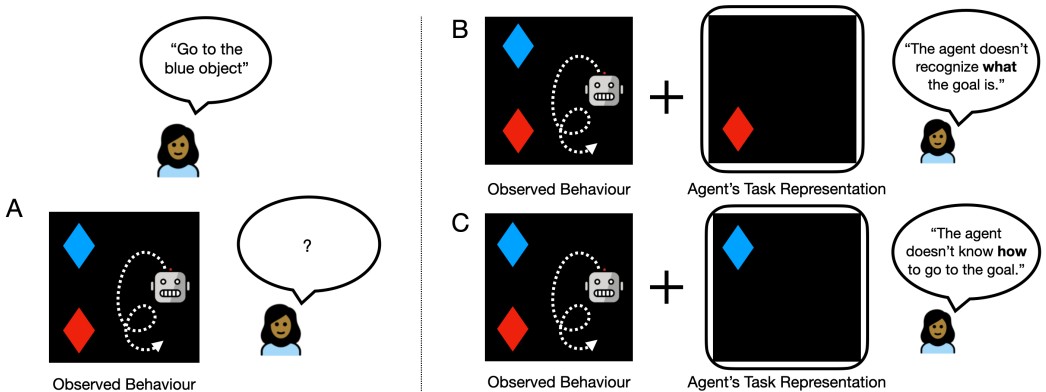

Figure 1: A human user trying to diagnose the agent's failure mode. (A) When the human only sees agent behaviour, it is ambiguous *why* it's failing. (B) If the human also has access to the agent's intermediate task representation for what it perceives to be relevant to the task, they can infer the type of error and thus parameters of the distribution shift. For example, if the agent is not attending to the target object, it is likely unfamiliar with the user's stated goal—i.e. a what error. (C) Alternatively, if the agent is attending to the target object but generates the wrong behaviour, this can indicate that it is unfamiliar with how to navigate to the object's location—i.e. a how error.

that only includes visual information relevant to solving the task. Our key insight is that while visual attention has been studied in the context of visualizing features of a deep learning model's black box predictions, *an incorrect visual mask can also help a human infer the underlying parameters of distribution shift in the event of a policy's failure*. This is done in the *feedback phase*, when we use the masked state to help a human infer whether the agent is attending to the right features but acting incorrectly (a how error) versus attending to the wrong features (a what error). To close the loop, we leverage the identified failure mode in the *adaptation phase* to perform more efficient finetuning via targeted data augmentation of the shifted parameter.

We formalize the problem setting and describe the underlying assumptions. Next, we present our interactive learning framework for diagnosing and fixing parameter-level shifts using human feedback. Through human experiments, we verify our hypothesis that visual attention is a more informative way for humans to understand agent failures compared to behaviour alone. Finally, we show that this feedback can be empirically leveraged to improve policy adaptation via targeted data augmentation. We call the full interactive training protocol the visual attention mask policy (VAMP).

## 2 RELATED WORK

**Goal-Conditioned Imitation Learning.** The learning technique used in our paper is goal-conditioned imitation learning (IM), which seeks to learn a multi-task policy end-to-end by supervised learning or "cloning" from expert trajectories (Abbeel & Ng, 2004; Ng et al., 2000; Ding et al., 2019). The learning from demonstrations framework means that we can optimize a policy without the need for a reward function (Pomerleau, 1988), albeit we cannot generate new behaviours without feedback. Moreover, unlike standard IM or IRL methods, goal-conditioned policies are capable of learning a single policy to perform many tasks. Unfortunately, generating enough expert demonstrations to cover a large test distribution is difficult.(Ziebart et al., 2008; Finn et al., 2016).

**Human-in-the-loop RL.** Interactively querying humans for data to aid in downstream task learning belongs to a class of problems referred to as human-in-the-loop RL (Abel et al., 2017; Zhang et al., 2019). Existing frameworks like TAMER (Knox & Stone, 2008) and COACH (MacGlashan et al., 2017) use human feedback to train policies, but are restricted to binary or scalar labeled rewards. A different line of work seeks to learn tasks using human preferences, oftentimes asking them to compare or rank trajectory snippets (Christiano et al., 2017; Brown et al., 2020). Yet another direction focuses on how to perform active learning from human teachers, where the emphasis is on gener-

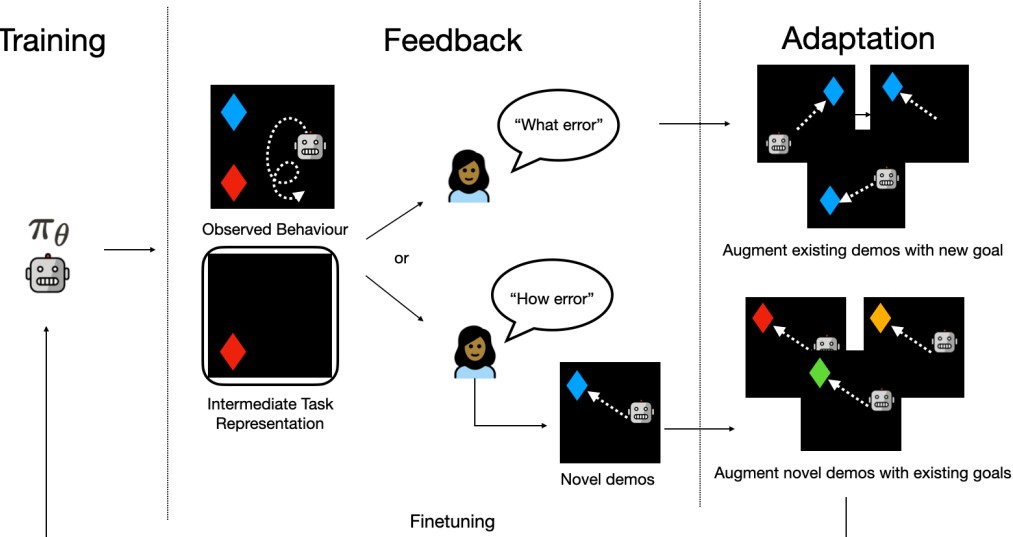

Figure 2: The framework overview. In the *training phase*, a policy learns to generate both actions as well as an intermediate task representation to help the human infer the failure mode in the *feedback phase*. We then leverage this information to perform targeted augmentation in the *adaptation phase*.

ating actions or queries that are maximally informative for the human to label (Bobu et al., 2022; Chao et al., 2010). The challenge with these approaches is that the feedback asked of the human is often overfit to specific failures or desired data points, and rarely scale well relative to human effort.

**Human-Interpretable RL** Ensuring that deep learning agents are intelligible to various concerns encompassing ethical, legal, safety, or usability viewpoints is a key focus of real-world deployments (García & Fernández, 2015). Methods range from training a policy to generate post-hoc explanations (Glanois et al., 2021), text-based descriptions of state predicates (Hayes & Shah, 2017), and Jacobian saliency maps (Greydanus et al., 2018). However, leveraging interpretable sequential decision-making systems to extract useful human feedback for adaptation has been under-explored.

## 3 PROBLEM FORMULATION

Our goal is to *develop a framework to help a human provide feedback about behaviour-invariant vs. dependent parameters of a distribution shift*. We formulate the problem as an imitation learning framework. In a goal-conditioned setting, goals are represented as $\mathcal{G}$. We consider environments represented by a Markov Decision Process (MDP) (Puterman, 2014) defined by tuple $M = \langle \mathcal{S}, \mathcal{A}, \mathcal{P}, \mathcal{R} \rangle$, where $\mathcal{S}$ is the state space, $\mathcal{A}$ the action space, $\mathcal{P} : \mathcal{S} \times \mathcal{A} \times \mathcal{S} \to [0, 1]$ the transition probability distribution, and $\mathcal{R} : \mathcal{S} \times \mathcal{G} \times \mathcal{A} \to \mathbb{R}$ the reward function. A parameterized policy is denoted as $\pi_\theta : \mathcal{S} \times \mathcal{G} \to \mathcal{A}$. In imitation learning (IM), rather than seeking to learn $\pi_\theta$ through interacting with the environment as in RL, we instead assume access to a set of expert demonstrations $\tau : \{(s_0^i, a_0^i, s_1^i, a_1^i, ... s_t^i, a_t^i)\}_{i=0}^n$ from which we "clone" $\pi_\theta$ from (Pomerleau, 1988). This yields a dataset of $n$ state-action-goal tuples $(s_t^i, a_t^i, g^i)$ for learning. While $g \in \mathcal{S}$ in typical settings, it need not be as long as we have a way of mapping the goal into a form that $\pi_\theta$ can process. Motivated with the idea that natural language is a flexible, intuitive interface for humans to communicate, we specify $g$ via a natural language instruction, resulting in the following standard GCBC loss:

$$\mathcal{L}_{\text{GCBC}}(\theta) = \mathbb{E}_{(s_t^i, a_t^i, g^i) \sim D_{\text{train}}}[\ell(\pi_\theta(s_t^i, g^i), a_t^i)] \tag{1}$$

where $\ell$ is mean squared error (MSE) for continuous actions and cross-entropy for discrete actions.

### 3.1 OUT-OF-DISTRIBUTION FAILURES

Like all supervised learning methods, GCBC suffers from *distribution shift* (Ross et al., 2011): i.e. $\pi_\theta$ could behave arbitrarily if faced with input at test that it did not see at training. Consider the

simplest case where the training set consists of states concatenated with goals, resulting in $D_{\text{train}}$: $[s, g] \in \mathcal{S} \times \mathcal{G}$. We can similarly define $D_{\text{test}}$: $[s', g'] \in \mathcal{S}' \times \mathcal{G}'$. Because we cannot guarantee that either the environment layout or the desired goal will remain unchanged at deployment, i.e. $[s', g'] \notin \mathcal{S} \times \mathcal{G}$, we face two possible sources of distribution shift: 1) the policy can fail to infer goals that are OOD (*what* failure) or 2) the policy can fail to produce correct actions for a goal location that is OOD (*how* failure). Our hypothesis is that in both cases, the erroneous actions generated by the policy are uninformative for a human to classify the source of failure.

## 3.2 ADAPTATION FOR BEHAVIOUR-INVARIANT VS. DEPENDENT TASKS

The adaptation challenge we face is how to most efficiently generate data for finetuning a policy to perform well on task distribution $\mathcal{D}_{\text{test}}$ given knowledge of shifted parameters from $\mathcal{D}_{\text{train}}$. In the most naive scenario, methods that assume knowledge of $\mathcal{D}_{\text{test}}$ uniformly sample tasks for querying human demonstrations for. In the case of goal-conditioned policies for visual navigation, this would include new goals that are both behaviour-invariant, e.g. object colors or types, as well as those that are behaviour-dependent, e.g. locations. However, such a finetuning strategy is practically inscalable when attempting to cover a large $\mathcal{D}_{\text{test}}$, where all tasks irrespective of which parameter has shifted require novel demonstrations (states and actions) from a human. Our insight is that rather than assuming demonstrations, i.e. new behaviours, are always required, we can *disentangle behaviour-dependent from behaviour-invariant tasks* and *only query for new data when needed*.

**Assumptions**. First, while $\mathcal{D}_{\text{train}} \neq \mathcal{D}_{\text{test}}$, we assume they are parameterized via a generative model capable of modifying visual parameters, i.e. the state space. These may include features such as object location and color and can be reasonably modified with access to a simulator or scene generator. Although we assume knowledge of the full parameter space, we do not know which parameters are shifting from $\mathcal{D}_{\text{train}}$ to $\mathcal{D}_{\text{test}}$ and therefore, which tasks require new actions. Thus, we assume the ability to query for this along with novel demonstrations from a human. Lastly, in order to "surgically" leverage existing data, we assume access to the original training demonstrations.

Given knowledge of the shifted parameter, we can perform targeted data augmentation. Consider the two failure types in Figure 2. If feedback is given that the shifted parameter is behaviour-invariant, e.g. a new goal object (A), we can augment our existing demonstrations with the desired goal via our generative model (if we know how to navigate to a blue key in the room we can also navigate to a red key in the same room). If instead feedback is given that the shifted parameter is behaviour-dependent, e.g. a new goal location (B), we can query for new actions from our human but then augment existing behaviour-invariant goals (if we are shown how to navigate to a blue key in a new room, we now also know how to navigate to a red key in that room). To deploy such "surgical" data augmentation techniques, we must develop a framework capable of extracting the shifted parameter from a human in a reliable and non-cumbersome manner. We next detail a framework for doing so.

## 4 THE VISUAL ATTENTION MASK POLICY LEARNING FRAMEWORK

We propose an interactive learning framework that leverages human feedback to disentangle behaviour-invariant from behaviour-dependent tasks. To do so, we assume conditionally independent parameters of our training distribution $\mathcal{D}_{\text{train}}$ (tasks that are learned by our policy at training time) and test distribution $\mathcal{D}_{\text{test}}$ (tasks that are desired at test time). Parameters range from goal object and color as well as location. Our framework is comprised of three phases: *training*, *feedback*, and *adaptation*. In the training phase, we train an end-to-end policy capable of generating an intermediate task representation, a masked state, as an additional human-interpretable output. In the feedback phase, we then use the visual mask to help a human diagnose the shifted parameter. Lastly, we leverage that feedback in the adaptation phase to perform targeted data augmentation.

## 4.1 TRAINING PHASE

As described in Section 3, the model for training policies capable of producing goal-conditioned attention masks are conditioned on language instruction $g$ and the current state $s$. Because our policy combines the standard GCBC loss in conjunction with an intermediate mask loss acting as a regularizer, we refer to our model as visual attention mask policy (VAMP). As a policy network, the following two modules are trained end-to-end: 1) attention module: processes the output of a pre-

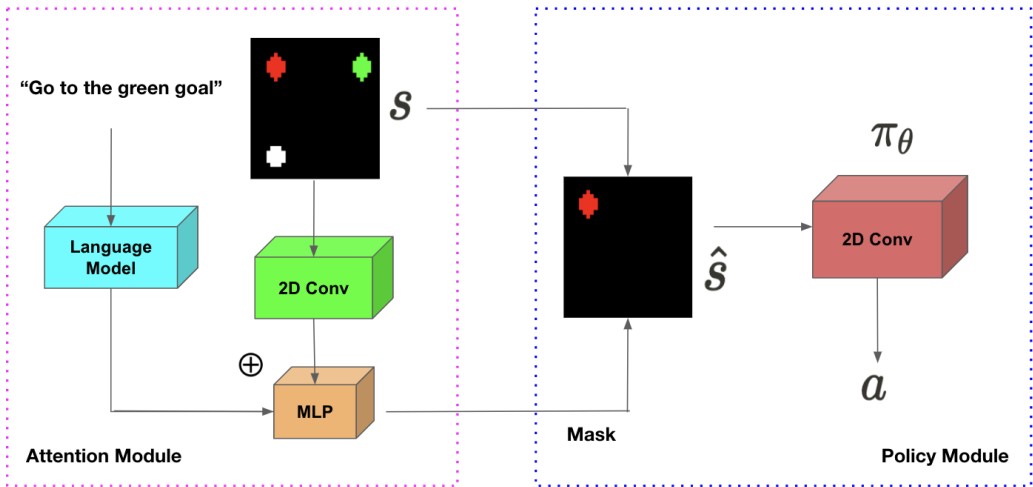

Figure 3: The architecture of our VAMP model. The attention module processes the goal specification and image state $s$. The resulting output is processed by the policy module along with the original state to create masked state $\hat{s}$, which is used for action prediction.

trained language model that turns natural language instruction $g$ into an embedding, concatenated with the output of a 2D ConvNet which takes state $s$ and produces a 1D vector; 2) policy module: a 2D ConvNet that takes the output of the attention module and combines it with state $s$ to produce masked state $\hat{s}$, which is used for action prediction. Note, the use of visual attention to focus on task-relevant features is well studied in the machine learning (Xu et al., 2015; Hayes & Shah, 2017) and cognitive science (Lindsay, 2020; Ho & Griffiths, 2022) literatures.

**Attention module:** Our attention module is comprised of three components: 1) a pre-trained general purpose language model called SentenceTransformer all-MiniLM-L6-v2 (Reimers & Gurevych (2019)), which we use to process a goal specification $g$ into an embedding, a vector of size 384; 2) a 4-layer 2D ConvNet with flattened last layer that we process image state $s$; and 3) a 2-layer MLP that we process the concatenated output into a mask.

**Policy module:** Our policy module processes the mask generated by the attention module with the original state $s$ to create a masked state $\hat{s}$ via a 4-layer 2D ConvNet to produce action $a$. The objective is:

$$\mathcal{L}_{\text{VAMP}}(\theta) = \mathbb{E}_{(s_t^i, a_t^i, g^i) \sim D_{\text{train}}} [\ell(\pi_\theta(s_t^i, g^i), a_t^i) + \beta \|\hat{s}^i\|_1] \tag{2}$$

As shown by the second term, our attention mask serves as a regularizer to reduce the information flow from the full state $s$ to the minimal state $\hat{s}$ that is required for action prediction, i.e. we wish to find the minimal representation of $s$ that is capable of correct action prediction. We additionally introduce weight $\beta$ to control the regularizer's contribution to the total loss so that we can scale the intensity of the masking. Full architecture and training details can be found in the appendix.

## 4.2 FEEDBACK PHASE

In the feedback phase, we now provide both the policy's output (actions) as well as its intermediate task representation to a human user for help in diagnosing specific policy failures. We generate distribution shifts of two types: 1) testing on goals outside of the training distribution (*what* or behaviour-invariant failures) and 2) testing on environment layouts with object locations, i.e. actions, outside of the training distribution (*how* or behaviour-dependent failures). For each error type, we create trajectories by sampling tasks from the test distribution and generating their corresponding intermediate task respresentation to show our human participants. We then query for feedback regarding which failure type is exhibited.

### 4.3 ADAPTATION PHASE

As shown in Figure 2, we now leverage the identified failure mode to perform targeted data augmentation of training demonstrations for our shifted parameter. For *what* or behaviour-invariant failures, we require only access to the generative model for modifying state space parameters, e.g. goal objects or colors. For *how* or behaviour-dependent failures, we require both the generative model as well as the ability to query for novel demonstrations, e.g. navigating to a new goal location.

**Finetuning a *what* failure**. We augment $D_{\text{train}}$ by replacing the original goal specification $g$ as well as the shifted parameters in the state (for example, if we wish to navigate to new keys, we augment demonstrations by replacing all existing goal objects with keys). We then finetune our policy using this dataset. This strategy requires no novel demonstrations since we are re-using behaviour-invariant demonstrations (i.e. modified states with existing actions).

**Finetuning a *how* failure**. Because we require new action sequences to navigate to a new goal location, we are required to collect novel demonstrations, i.e. states and behaviours. While these demonstrations would typically come from the human user, we sample from a simulated expert for ease of exposition. We first sample $g \sim \mathcal{D}_{\text{test}}$ and collect expert demonstrations $D_{\text{novel}}$ for reaching $g$. We then leverage knowledge of the behaviour-invariant parameters (e.g. goal object or color) $\sim \mathcal{D}_{\text{train}}$ to create an augmented dataset of these demonstrations for finetuning.

## 5 EXPERIMENTS

In this section, we detail a set of empirical results to answer the following questions: 1) Do intermediate task representations help humans diagnose the underlying parameter shift in the event of a policy failure? 2) Does correctly leveraging this information for data augmentation improve downstream policy performance? 3) Do these gains hold when evaluated on real human subjects?

To answer these questions, we evaluate our proposed framework on two domains: a single-goal navigation task and a multi-room compositional task. For each domain, we first detail the task generation process for the training and test distributions. We then conduct experiments with a simulated human oracle to explore the maximum performance gain that can be achieved by an optimal finetuning strategy with a fixed data labeling budget. Lastly, we verify these results with real human feedback collected from user studies on diagnosing distribution shifts.

### 5.1 MULTI-OBJECT NAVIGATION

We create an image-based environment where an agent is tasked with navigating to a goal object of one color while ignoring a distractor object of a different color. States are fully-observable RGB images of dimension $36 \times 36 \times 3$ and the action space is continuous and represents the $(x, y)$ distance that the agent can move in 2D space. The environment was created to test a simple object navigation domain while preserving a continuous action space of higher complexity relative to discrete gridworld tasks.

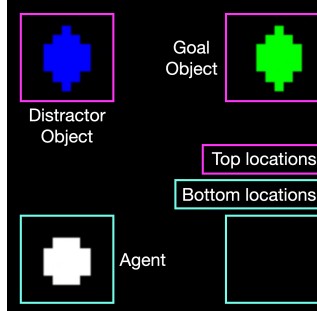

Figure 5: An example multi-object navigation task for "go to the green object".

The environment contains 4 colors (red, green, blue, yellow) and 4 starting locations (grid corners). A task distribution $\mathcal{D}$ specifies the generative parameters for sampling the goal location/color and can contain any subset of the environment parameters above. To generate tasks, we first place a goal object of color and location uniformly sampled from $p(\{\text{colors}\}))$ and $p(\{\text{locations}\}))$. We then place a distractor object and the agent at two randomly sampled remaining locations. Lastly, we assign the distractor object a randomly sampled unique color (the agent is always white). The task specification is defined via a language instruction "go to the `<goal color>` object".

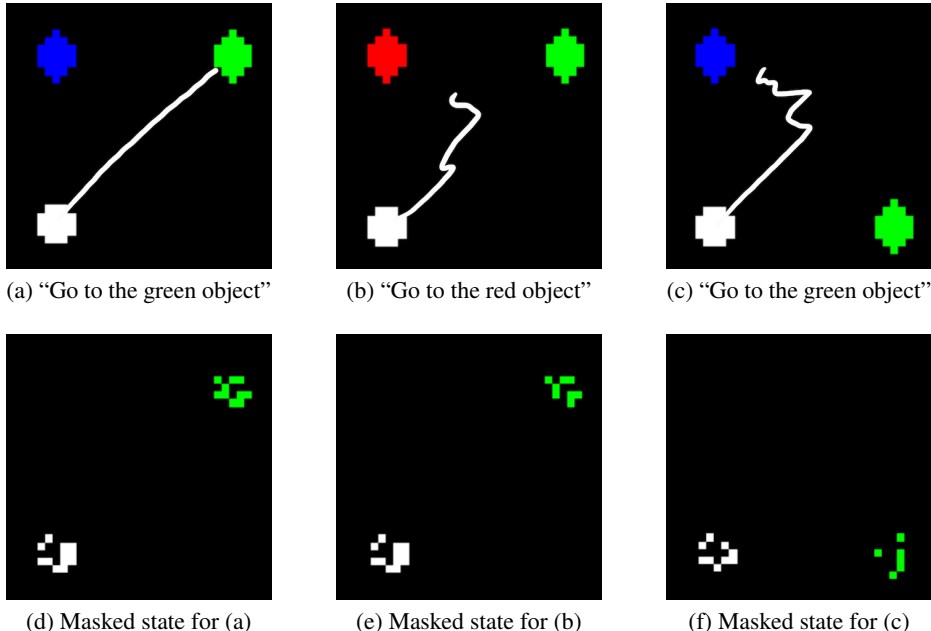

(a) "Go to the green object"    (b) "Go to the red object"    (c) "Go to the green object"

(d) Masked state for (a)    (e) Masked state for (b)    (f) Masked state for (c)

Figure 4: Examples of the multi-object navigation task. An agent (white) trained to navigate to green objects in all locations, when asked to navigate to a green object, will produce successful actions (a) as well as a task representation attending to the correct goal (d). If the agent is asked to instead navigate to a red object (b), it will exhibit a *what* failure, with the mask attending to the wrong object (e). This is in contrast to a *how* failure, where an agent trained to navigate to objects on the top will fail to navigate to objects on the bottom (c), even though the mask attended to the correct goal (f).

For each $\mathcal{D}_{\text{train}}$, we generate 100 tasks. For each task, we generate an expert demonstration of length 20 by taking continuous actions from the agent's starting location to the goal object, yielding 100 demonstrations for our initial policy training. We train and test on 2 possible distribution shifts.

## 5.2 MULTI-ROOM COMPOSITIONAL TASK

We also design a task with a multi-room multi-task compositional structure to explore how our method scales with task compositionality and long-horizon goals. We adapt the DoorKey environment from Minigrid (Chevalier-Boisvert et al. (2018)) and create an environment composed of three sub-tasks (pick up a key, use the key to unlock a door, then navigate through the door to a goal). The state space is fully-observable and consists of RGB images of dimension $36 \times 36 \times 3$. The action space is size 6 and allows for cardinal movements, picking up/dropping a key, and using a key to open a door.

There are 4 possible colors (red, green, blue, yellow), and 10 key, door, and goal locations each. $\mathcal{D}$ contains parameters for sampling each sub-task color and location. We generate tasks by first uniformly sampling a key color/location, door color/location, and goal color/location from $\mathcal{D}$. We next place three lava objects as immovable obstacles at a randomly sampled location for each sub-task. The agent always begins at the top left corner. The task specification is defined via a language instruction "go to the `<key color>` key, `<door color>` door, `<goal color>` goal".

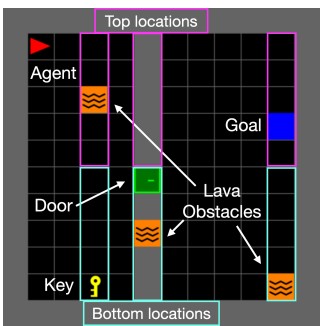

Figure 6: An example multi-room compositional task for "go to the yellow key, green door, blue goal".

To explore how task compositionality impacts finetuning efficiency, we create distributions for each sub-task by varying the parameters for one object while holding the

others constant (for example, a distribution with randomly sampled red keys in all locations would hold the door and goal colors/locations constant). For *what* shifts, we train on red colors for each object and test on green, blue, and yellow colors in all locations. For *how* shifts, we train on objects in the top half of the room and test on the bottom half. We generate 6 unique shifts while maintaining the compositional structure of the task. We sample 100 tasks from each $\mathcal{D}_{\text{train}}$ and for each generate an expert demonstration of the agent successfully completing all 3 sub-tasks.

## 5.3 HUMAN EXPERIMENTS

For the finetuning strategies which require human feedback, we conduct user studies at [Anonymous Institution]. We recruit 12 subjects for each domain (71% male, age 18-31). 88% of participants attested to having a technical background, although only 17% have worked with machine learning.

The user study is comprised of two phases: a familiarization and feedback phase. In the familiarization phase, we introduce the user to the task context, environment, and an example of each failure type. In the feedback phase, we first show the user the agent's behaviour on the test set (5 trajectories for each shift in the multi-object navigation task and 2 trajectories for each shift in the multi-room compositional task, randomized). We then ask for feedback on whether they believed the failure was due to a *what*, *how*, or *unclear* failure. Lastly, we then show the users the same trajectories with their corresponding attention mask as additional information, and request a second round of feedback. Altogether, we received 120 data points for each control and experimental group per domain. For each participant, a finetuning strategy was selected based on the average predicted accuracy of each failure type (e.g., a participant that predicted 4 out of 5 *how* failures incorrectly as a *what* failure would deploy the wrong strategy). *Unclear* responses were treated as incorrect.

| Domain | Distribution Shift | | | Correct Feedback (%) | |
|--------|--------------------|--------|--------|---------|------|
| | $D_{\text{train}}$ | $D_{\text{test}}$ | Failure | Naive H | Ours |
| Multi-Obj | RG goals, all loc | BY goals, all loc | *what* | 6.7 | **78.3** |
| | all goals, bottom loc | all goals, top loc | *how* | 18.3 | **95.0** |
| Multi-Room | R keys, all loc | GBY keys, all loc | *what* | 4.2 | **83.3** |
| | all keys, top loc | all keys, bottom loc | *how* | 8.3 | **75.0** |
| | R door, all loc | GBY door, all loc | *what* | 16.7 | **70.8** |
| | all doors, top loc | all doors, bottom loc | *how* | 4.2 | **54.2** |
| | R goals, all loc | GBY goal, all loc | *what* | 50.0 | **70.8** |
| | all goals, top loc | all goals, bottom loc | *how* | 0.0 | **62.5** |

Table 1: Rate of correct human responses for each distribution shift. We see that our method enables human users to more accurately diagnose the underlying distribution failure type.

## 5.4 EVALUATION

For each domain, we assess performance of the final policy on 20 sampled tasks from $D_{\text{test}}$ after finetuning with the selected strategy. The metric of evaluation that we seek to minimize is *user effort*, which we measure by the number of novel demonstrations that are required for any particular strategy. While we generate these demonstrations here, this is motivated by the idea that in a deployment scenario, this cost is largely incurred by the human teacher who must provide novel demonstrations, and not from data augmentation or finetuning performed in the factory. We make 5 comparisons: 1) no finetuning, 2) finetuning with no human feedback, 3) finetuning with naive human feedback, 4) finetuning with informed human feedback (our method), and 5) oracle finetuning.

**No finetuning (None)**: No finetuning of the policy is permitted.
**Finetuning with no human feedback (No Human)**: Our second baseline describes the scenario where we receive no human feedback and therefore receive no information about the desired test distribution. In this case, we allow the generation of 20 novel tasks by randomly sampling from all possible environment parameters, creating their corresponding demonstrations, and finetuning.
**Finetuning with naive human feedback (Naive Human)**: Our third comparison is the case where we receive human feedback without the intermediate task representations from our method. This

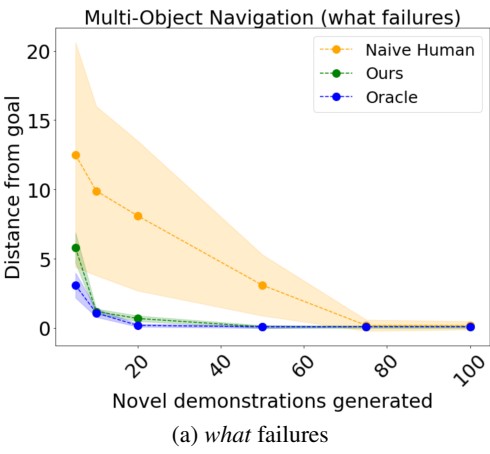
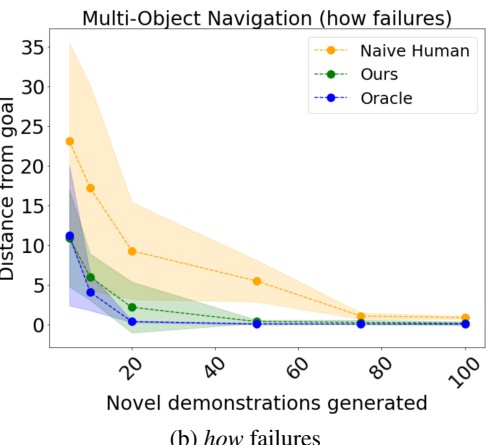

(a) *what* failures            (b) *how* failures

Figure 7: Final policy performance as a function of user effort (measured by number of novel demonstrations generated) on the multi-object navigation task. We see that our method requires less user effort to achieve higher policy performance relative to naive human feedback.

represents our control group, where human participants are shown agent trajectories only. For every human participant, we finetune a policy according to their selected (potentially erroneous) strategy and average policy performance across participants. If a *how* error was selected, we allow the generation of 20 novel demonstrations prior to augmentation (0 are required for a *what* strategy).
**Finetuning with informed human feedback (Ours)**: Our fourth comparison is our method, where human feedback is informed by both agent trajectories as well as intermediate task representations. We finetune similarly to Naive Human above, but leverage a more accurate augmentation strategy.
**Finetuning with perfect feedback (Oracle)**: For completion, we also include a comparison to the maximum adaptation gain possible if we selected the strategy that perfectly predicted the correct failure mode. The same fixed budget for maximum 20 novel demonstrations are applied here.

| Domain | Distribution Shift | | | Distance from Goal (std) | | | | |
|---|---|---|---|---|---|---|---|---|
| | $D_{\text{train}}$ | $D_{\text{test}}$ | Failure | None | No H | Naive H | Ours | Oracle |
| Multi-Obj | RG goals, all loc | BY goals, all loc | *what* | 13.7 (4.6) | 11.5 (4.0) | 8.1 (5.4) | **0.7 (0.2)** | 0.2 (0.1) |
| | all goals, bottom loc | all goals, top loc | *how* | 21.7 (9.1) | 16.1 (6.2) | 9.3 (6.1) | **2.2 (3.2)** | 0.4 (0.1) |
| Multi-Room | R keys, all loc | GBY keys, all loc | *what* | 10.2 (2.4) | 6.2 (0.7) | 11.7 (2.1) | **0.3 (0.6)** | 0.4 (0.8) |
| | all keys, top loc | all keys, bottom loc | *how* | 12.3 (0.1) | 7.1 (1.3) | 9.5 (5.4) | **3.6 (2.2)** | 1.7 (1.4) |
| | R doors, all loc | GBY doors, all loc | *what* | 8.0 (1.9) | 5.2 (1.6) | 5.5 (0.8) | **2.8 (1.9)** | 1.0 (0.3) |
| | all doors, top loc | all doors, bottom loc | *how* | 9.1 (2.1) | 4.6 (1.1) | 6.1 (2.2) | **4.0 (0.7)** | 0.7 (0.9) |
| | R goals, all loc | GBY goals, all loc | *what* | 6.2 (1.9) | 4.7 (1.9) | 8.9 (1.8) | **0.7 (0.4)** | 0.2 (0.4) |
| | all goals, top loc | all goals, bottom loc | *how* | 8.1 (0.9) | 3.7 (0.7) | 6.1 (2.3) | **0.3 (0.7)** | 0.3 (0.6) |

Table 2: Final policy performance for all tested distribution shifts in both domains. Our method outperforms all baselines, and rivals perfect feedback from an oracle in some cases.

Table 1 shows that human feedback using our framework results in more accurate distribution shift diagnosis vis-a-vis those without. Table 2 shows the results of downstream policy performance on each distribution shift after finetuning with a maximum budget of 20 novel demonstrations per strategy. This empirically demonstrates how more accurate feedback results in improved policy performance given a fixed quota of user effort. Note, a naive human is sometimes outperformed by no human at all, highlighting that an incorrect data augmentation strategy can sometimes be worse than random sampling demonstrations from the test distribution.

# 6 DISCUSSION AND CONCLUSION

**Summary.** We leveraged existing techniques to create a human-in-the-loop framework for diagnosing and fixing distribution shift in end-to-end sequential systems. We showed that our framework

effectively utilizes insights from cognitive science to produce intermediate task representations capable of aiding humans in diagnosing underlying distribution shifts. We also demonstrated the empirical performance benefit of our method in reducing human effort for downstream adaptation.

**Limitations.** There are parameters that are difficult to practically disentangle through visual attention alone such as object occlusion, shape and texture, and partially observed scenes. Moreover, we assume access to a generative model capable of manipulating those parameters, a challenging task without high-quality scene representation. We remain optimistic that advances in representation learning for feature disentanglement can be easily incorporated into our framework.

**Future Impact.** As human-in-the-loop systems are increasingly deployed, we must find more efficient ways of leveraging feedback for improving learning systems if we wish to practically adapt to user preferences. Moreover, if we have systems operating around and with humans, we must create more transparent, reliable methods of communicating the underlying parameters of *why* they fail.

## 7 ETHICS STATEMENT

Since our paper relies heavily on human experiments and data for evaluation, we attest to the following details related to human subject evaluation and data privacy. An institutional IRB was filed prior to subject recruitment. All human subjects were asked to voluntarily participate in the experiment of their own free will. While we gathered basic demographic information (age, gender, and technical background), participant information was anonymized for analysis and referred to only by ID number. We collected no further information.

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

# A APPENDIX

## A.1 TRAINING AND ARCHITECTURE DETAILS

**Architecture.**

The language model processes a goal sentence in natural language into an embedding of size 384, which we additionally process through a linear MLP of output size 100.

Our Conv2D blocks process both the image state $s$ as well as masked state $\hat{s}$ and consist of 3 stacked Conv2D layers of output channel sizes 32, 64, 32, kernel sizes 8, 4, 3 and strides 4, 2, and 1. Each output layer is processed by BatchNorm2D as well as a ReLU activation. After the last Conv2D layer, we flatten the output process through BatchNorm1D, then through a linear layer. The Conv2D block for processing the initial state $s$ generates an output of size 100 while the Conv2D block for processing masked state $\hat{s}$ directly generates an output of the action size.

Our MLP block consists of two-layer stacked MLP of input size 200, hidden size 100, and output size 1296, which we then reshape into a 1 channel image of dimension $36{\times}36$.

**Training.** We use Adam for optimization with a learning rate of 0.001. All training was performed in PyTorch with SGD. For policy training of tasks from $\mathcal{D}_{\text{train}}$, we train for 1200 epochs using a batch size of 50. For finetuning, we train the policy for an existing 1200 epochs. We performed a parameter sweep over the regularizer term $\beta$ and used 0.005 for our experiments.

## A.2 MULTI-OBJECT NAVIGATION

As the multi-object navigation task is one that is created specifically for our project, we detail the full generation details below.

Formally, the environment is characterized by a tuple $\langle \Lambda, \Omega \rangle$ where $\Lambda$ is the agent and $\Omega$ is the set of objects in the room. For each task distribution, we define a finite set of Colors := {red, green, blue, yellow} and two-dimensional Locations := $\{(-5, -5), (-5, 5), (5, -5), (5, 5)\}$. At reset, $\Lambda$ is placed at location $\lambda_\Lambda \sim U(\text{Locations})$. Two objects, $\omega_1$ and $\omega_2$, are generated, each with a color $c_n$ and a location $\lambda_n$, with $c_1, c_2 \sim U\{\text{Colors}\}$, $\lambda_1 \sim U[\{\text{Locations}\} \setminus \{\lambda_\Lambda\}]$ and $\lambda_2 \sim U[\{\text{Locations}\} \setminus \{\lambda_\Lambda, \lambda_1\}]$. We define an environment configuration by a random variable $X = (x_{\lambda_\Lambda}, x_{c_1}, x_{\lambda_1}, x_{c_2}, x_{\lambda_2})$, where a specific configuration is characterized by the vector $x = (\Lambda_\lambda, c_1, \lambda_1, c_2, \lambda_2)$.

We sample a goal $\omega_g \sim U\{\omega_1, \omega_2\}$ and assign the remaining object as distractor $\omega_d$. A language generation function $\psi : w_g \to \mathcal{L}$ can be thought of as producing the goal specification i.e. language instruction that a human would provide at deployment. An image generation function $\phi : \Omega \times X \to \mathcal{S}$ renders vector $x$ into an image state $s$, where the agent is a white circle, the goal a diamond of its assigned color, and the distractor a diamond of its assigned color.

A task distribution $\mathcal{D}$ contains the parameters of the generation function (colors and locations of the agent, goal, and distractor). For example, one such distribution could be {red, green} and {(-5,-5),(-5,5)}. A specific task $\mathcal{D}_i$ generates an environment as specified above, yielding a tuple $\langle \psi(\omega_g), \phi(\omega_g, x_0) \rangle = \langle L, s_0 \rangle$ and describes a specific image and its associated goal specification. We sample 100 tasks for each training distribution $\mathcal{D}_{\text{train}}$ and generate each an expert trajectory $d : (s_0, a_0, \ldots s_{20}, a_{20})$ by taking continuous actions toward the goal object, yielding 100 trajectories which we use to train our GCBC policy.

