# OpenReview forum: "Fast Adaptation via Human Diagnosis of Task Distribution Shift"
_ICLR.cc/2023/Conference — Submitted to ICLR 2023_

### Official Review · Reviewer_i8yJ · 2022-10-24

**Confidence:** 4
**Correctness:** 2
**Technical Novelty And Significance:** 3
**Empirical Novelty And Significance:** 2
**Recommendation:** 5

**Clarity, Quality, Novelty And Reproducibility:**

The paper is clearly written and easy to follow. To facilitate model performance, the authors adopted attention visualization in a goal-conditioned imitation learning task to include human diagnosis. However, its novelty and significance are limited by the current model design and experimental setting. Given the implementation details, the experiments seem reproducible.

**Strength And Weaknesses:**

[+] This paper studies the important interactive learning problem and propose a pipeline to include human in the loop for improving model performance under task distributional shift by adding a visualizable intermediate representation that could be used for diagnosis.

[+] In the 2D navigation and goal-conditioned reinforcement learning setting, the proposed baseline is simple and effective where the attention visualization can provide clear clues on what type of errors the model made under a shifted test distribution.

[-] One major concern of this paper lies in the significance and generalizability of the proposed method. Although the 2D navigation task featured in this paper is a good and clear example to illustrate the idea, it is not clear how to add a visualizable or human-interpretable intermediate representation for other general tasks. This makes the current design's contribution somewhat limited as the fast adaptation relies heavily on human diagnosis and efficiently augmenting learning data. Therefore, I do not see its potential as a general plug-and-play module for solving other tasks as well.

[-] As the authors stated, the use of visual attention to focus on task-relevant features is well-studied and widely adopted. However, it also relies on an interpretable canvas for visualization (image, video, or text) and gradually lose its explainability when the model size or depth increases. This adds difficulty to the current design for generating diagnosing visualizations in complex tasks with more computation-heavy (e.g., deep transformers) models.

[-] The current evaluation results are mainly ablative studies of the VAMP model without proper comparison with baselines. Following the previous two points, the authors should consider comparing state-of-the-art models with the proposed VAMP to justify the difficulty and challenge of the proposed 2D navigation problem since if such models can already perform well, the point of using this experimental setting should be questioned.

**Summary Of The Paper:**

This paper proposes to leverage visual attention as a diagnosis tool for reflecting the reason for error during models' task execution. The authors leverage a 2D language navigation task as the test bed and tested mainly using goal-conditioned imitation learning methods. They visualize attention on the 2D map to show whether the model has a wrong understanding of the goal or can not reach the goal. Humans can diagnose the failure of models based on this visualization to augment learning data sources to facilitate task execution. The authors showed that their visualization can help humans diagnose problems of models and further improve the navigation task in cases where train and test distributions shift.

**Summary Of The Review:**

This paper proposes to include human diagnosis in 2D navigation by adding additional visualization of goal localizations in the policy module for adaptation to task distribution shifts. However, the benefits brought by the method rely heavily on its design and it's unclear how to generalize the method proposed to more complex tasks. Therefore, I don't think it meets the bar for acceptance for now. I suggest the authors should justify that they are solving a challenging task and the proposed model could be leveraged in more complex scenarios in the following revisions.

---

> ### Author Response · Authors · 2022-11-13
> **Author Response to Reviewer i8yJ**
>
> We thank the reviewer for the time spent in providing feedback. Please refer to the general response for answers regarding generalizability of the framework to additional tasks.
>
> >**Q1**: As the authors stated, the use of visual attention to focus on task-relevant features is well-studied and widely adopted. However, it also relies on an interpretable canvas for visualization (image, video, or text) and gradually lose its explainability when the model size or depth increases. This adds difficulty to the current design for generating diagnosing visualizations in complex tasks with more computation-heavy (e.g., deep transformers) models.
>
> **A1**: We thank the reviewer for acknowledging the widespread use and adoption of visual attention as a method for communicating to humans. In fact, the very notion that natural images remain an aligned source of raw sensory input that both humans and robots perceive the world in is a major motivator for the diagnostic tool instantiated in our framework! However, we are not aware of literature detailing the failure of visual attention to visualize input spaces given larger models (in fact, we are only aware of literature where deep nets are capable of producing relatively high fidelity segmentation masks via attention [1]). Moreover, an interesting finding of our framework is that attention, *even if incorrect*, is still able to provide a usable signal to the human regarding the error type that is experienced by the model because it highlights inconsistencies between behaviour-invariant vs. behaviour-dependent tasks. Would the reviewer be so kind as to direct us to the relevant work regarding the inability to train visual attention masks in deeper/larger models?
>
> >**Q2**: The current evaluation results are mainly ablative studies of the VAMP model without proper comparison with baselines. Following the previous two points, the authors should consider comparing state-of-the-art models with the proposed VAMP to justify the difficulty and challenge of the proposed 2D navigation problem since if such models can already perform well, the point of using this experimental setting should be questioned.
>
>
> **A2**: While important, we believe the question of “will larger models solve this task” to be orthogonal to the problems addressed in our paper. While it remains certain that larger, deeper networks will be continually leveraged across all robotic domains, we are addressing the question of “what’s the best way to identify an error in a model and fix it once it has failed?”, a challenge that even state of the art models face under distribution shift [2-6], which is the scenario considered under our experiments. Given this axis of study, we believe the baselines captured in our paper meaningfully represent the current state of the art strategies for querying data from humans for policy finetuning (including naive sampling when either informed or not informed by the test distribution, and assuming access to a human demonstrator or not). We would be happy to consider alternative baselines related to better policy finetuning strategies if the reviewer would be so kind as to provide them.
>
> **References**
>
> 1. He, Kaiming, et al. "Mask r-cnn." Proceedings of the IEEE international conference on computer vision. 2017.
> 2. Thrush, Tristan, et al. "Winoground: Probing Vision and Language Models for Visio-Linguistic Compositionality." Proceedings of the IEEE/CVF Conference on Computer Vision and Pattern Recognition. 2022.
> 3. Conwell, Colin, and Tomer Ullman. "Testing relational understanding in text-guided image generation." arXiv preprint arXiv:2208.00005 (2022).
> 4. Sagawa, Shiori, et al. "Extending the wilds benchmark for unsupervised adaptation." arXiv preprint arXiv:2112.05090 (2021).
> 5. Koh, Pang Wei, et al. "Wilds: A benchmark of in-the-wild distribution shifts." International Conference on Machine Learning. PMLR, 2021.
> 6. Miller, John, et al. "The effect of natural distribution shift on question answering models." International Conference on Machine Learning. PMLR, 2020.

---

> > ### Comment · Reviewer_i8yJ · 2022-12-01
> > **Post-rebuttal response**
> >
> > Thank the authors for the response and clarifications. For the attention mask issue, I was originally expressing my concerns over more complex tasks where attention visualizations are not that helpful for diagnosis. This is combined with the use of large models where the intermediate representations do not necessarily preserve any semantic meaning (which is not about whether attention mask could be trained, but about the fact that what these attention maps are over at will be hard to define for them to be semantically meaningful). Finally, I also share the feeling of other reviewers that the current response is not convincing enough for justifying its potential as a general learning framework. Therefore, I raised my score to weak reject hoping that the authors could address these concerns in later revisions.

---

### Official Review · Reviewer_vMKq · 2022-10-28

**Confidence:** 3
**Correctness:** 3
**Technical Novelty And Significance:** 2
**Empirical Novelty And Significance:** 2
**Recommendation:** 5

**Clarity, Quality, Novelty And Reproducibility:**

The clarity of this work could be improved, as noted in the above box. Additional care needs to be taken so that the work is technically sound and precise.

The idea of using attention masks as an error analysis tool is not new. However, the idea of diagnosing the failure mode of a policy to fix it with data augmentation is indeed original.

**Strength And Weaknesses:**

## Strengths
The main strength of this work is that it studies and interesting and practical problem that is relevant to the RL community. It offers the new idea of diagnosing the failure mode of the distribution shift to update the policy via data augmentation.

## Weaknesses

The main weakness of this work is that it's unclear how general the method is and how well it scales to tasks beyond grid worlds. Specifically, while it seems quite possible that including attention masks as an error analysis tool is widely applicable, failure modes may be less interpretable and fall on axes beyond "what" and "how" when moving to 3d tasks with many steps. For these sorts of tasks, it may further be unclear what steps to take even after observing the attention mask. Additionally, the current setup requires fairly strong and restrictive assumptions. Specifically, the experiments assume the ability to generate new tasks of the same distribution shift type, which is unlikely to appear in practice. Further discussion of these aspects is warranted.

The experimental setup could be improved:
- The results in Table 1 seem somewhat suspicious. The "Naive H" feedback appears to be almost always incorrect. However, if you choose the opposite of what "Naive H" predicts, then this appears to be more accurate than the mask-augmented feedback much of the time. What's going on here?
- It would be helpful to tease apart the effects of including more demonstrations vs. including demonstrations of the correct type. For example, does increasing the number of demonstrations with oracle feedback further improve performance? What if you also added some demonstrations of the wrong type?
- Specifically, how are the demonstrations generated to target "what" and "how" types? I assume that the "No human" baseline uniform task generation is different than just adding half "what" and half "how" demonstrations. If so, it would be useful to include that as a baseline, which would probably perform better and requires no human intervention?

This work lacks clarity and precision in many areas. Below are a list of examples:
- The objective in Equation (2) is somewhat unclear because the parameters are not included in the objective. It's written under the policy module paragraph, which suggests that it's only an objective for the policy module, but the regularization term on $\hat{s}$ is independent of the policy module. Additionally, the regularization term needs to be inside the expectation, otherwise it's incorrect? It's also unclear how this "reduce(s) the information flow from the full state s to the minimal state." What information is flowing where?
- How are "unclear" responses factored into the accuracy of Table 1? Are they disregarded entirely? If so, what's the point of including such a response?
- Why are augmentations not included for the "what" distribution shifts? e.g., the text states: “If a how error was selected, we allow the generation of 20 novel demonstrations prior to augmentation (0 are required for a what strategy).”
- It's unclear what it means that the training and test splits are related by a generative model
- The problem formulation states that it encapsulates both discrete and continuous control, but the objective (1) is written only for continuous control. Specifically, the inside of the expectation should probably just be $\log \pi(a \mid s, g)$. Formally, the goal should be augmented into the state, or otherwise the reward function is undefined on the goal.

**Summary Of The Paper:**

This work considers the problem of diagnosing why a policy fails when presented with an out-of-distribution task. Specifically, this work creates a framework to determine if the policy fails for a "how" reason (e.g., the policy does not know how to accomplish the provided goal) or a "what" reason (e.g., the policy does not know what the goal is at all). Then, this information is used to fine-tune the policy on data augmentations specifically designed to combat the determined failure mode.

**Summary Of The Review:**

The idea of diagnosing and fixing policy behavior on out-of-distribution tasks is interesting and practically useful. However, the proposed method has several significant drawbacks, including its inability to generalize to more complex tasks of practical value. Hence, I cannot currently recommend acceptance.

---

> ### Author Response · Authors · 2022-11-13
> **Author Response to Reviewer vMKq**
>
> We thank the reviewer for the time spent in providing feedback. Please refer to the general response for answers to questions regarding generalizability of the framework to additional tasks, as well as an updated revision clarifying experimental details and a proposed revision for structural changes.
>
> >**Q1**: The results in Table 1 seem somewhat suspicious. The "Naive H" feedback appears to be almost always incorrect. However, if you choose the opposite of what "Naive H" predicts, then this appears to be more accurate than the mask-augmented feedback much of the time. What's going on here?
>
> **A1**: The reviewer has identified a very unclear experimental detail: when we stated “unclear responses were disregarded” above Table 1, we meant that they were not included in our calculation of “correct responses”, not that they were thrown out of the analysis. In fact, most (upwards of ~70% in several cases) of the human feedback we received for “Naive H” responses were “unclear”, and therefore not included in the count of “correct” responses. This is in contrast to a majority of respondents accurately identifying the error type for the “Informed H” condition. This has been clarified in our revision.
>
> >**Q2**: It would be helpful to tease apart the effects of including more demonstrations vs. including demonstrations of the correct type. For example, does increasing the number of demonstrations with oracle feedback further improve performance? What if you also added some demonstrations of the wrong type?
>
> **A2**: These are great suggestions, and those also empirically evaluated by us. In Figure 7 (p.9), we studied the effect of increasing different types of demonstrations on final performance. On the multi-object navigation task, we see that finetuning with oracle demonstrations requires less trajectories to stabilize to a better performing policy, while finetuning with Naive H (largely incorrect) takes far more to achieve the same performance (~20 vs. ~75 demonstrations). Finetuning with fully incorrect demonstrations (those that never achieve the desired task) do not improve the policy at all irrespective of number, and were omitted for clarity. We see the same trends on the multi-room composition tasks.
>
> >**Q3**: Specifically, how are the demonstrations generated to target "what" and "how" types? I assume that the "No human" baseline uniform task generation is different than just adding half "what" and half "how" demonstrations. If so, it would be useful to include that as a baseline, which would probably perform better and requires no human intervention?
>
> **A3**: Please refer to the first part of our general response.
>
> >**Q4**: The objective in Equation (2) is somewhat unclear because the parameters are not included in the objective...
>
> **A4**: The reviewer is correct - we have rewritten Equation 2 to include the regularization term inside the expectation and clarification of the policy parameters. The intuition behind the regularization term is to find the minimal representation of s that is required to correctly predict action a, i.e. we wish to extract a minimal state s_hat from s. These have been clarified in the uploaded revision.
>
> >**Q5**: How are "unclear" responses factored into the accuracy of Table 1? Are they disregarded entirely? If so, what's the point of including such a response?
>
> **A5**: We thank the reviewer for pointing out an unclear detail: when we stated “unclear responses were disregarded” above Table 1, we meant that they were not included in our calculation of “correct responses”, not that they were thrown out of the data entirely. We felt it important to include this option as an alternative for suggesting user uncertainty rather than forcing a random guess, which may have biased the true correct responses. This has been clarified in our updated revision.
>
> >**Q6**: Why are augmentations not included for the "what" distribution shifts? e.g., the text states: “If a how error was selected, we allow the generation of 20 novel demonstrations prior to augmentation (0 are required for a what strategy).” It's unclear what it means that the training and test splits are related by a generative model.
>
> **A6**: Please refer to the first section of our general response.
>
> >**Q7**: The problem formulation states that it encapsulates both discrete and continuous control, but the objective (1) is written only for continuous control...
>
> **A7**: These are great points, and we have re-written Equation 1 to clarify the loss for both continuous and discrete scenarios. Moreover, we have clarified our definition of the state and reward as depending on g. Please refer to the revised version.

---

> > ### Comment · Reviewer_vMKq · 2022-11-18
> > **Reviewer Response**
> >
> > I appreciate the authors' response and clarification, which has addressed many of my concerns about clarity. I have a few follow-up questions, though I understand that I am responding at the very end of the discussion period, and the authors may not have a chance to respond.
> >
> > > A5: We thank the reviewer for pointing out an unclear detail: when we stated “unclear responses were disregarded” above Table 1, we meant that they were not included in our calculation of “correct responses”, not that they were thrown out of the data entirely. We felt it important to include this option as an alternative for suggesting user uncertainty rather than forcing a random guess, which may have biased the true correct responses. This has been clarified in our updated revision.
> >
> > This seems potentially somewhat unfair, as it results in an option that is always not correct. Perhaps the users would have leaned toward the correct answer, but were unsure, and instead selected the "unclear" option instead.
> >
> > More significantly, while I appreciate the authors' response, my main concern of the scalability of the proposed method remains. Even if we assume the optimistic view on generative models that the authors purport in the rebuttal, which enables generating new demonstrations / tasks in a 3D setting, and further that failures can be visually diagnosed, the more pressing concern seems to be whether anything actionable can actually be done. In the 2D setting, where failure modes divide cleanly between "what" and "how" errors can relatively easily be addressed, but in more complex domains, even if we can see a particular failure, I have concerns that it remains unclear how to augment the data in order to address this failure in general. Given that this main concern still stands, I am inclined to keep my original score, though the authors' efforts are much acknowledged and appreciated.

---

> > > ### Author Response · Authors · 2022-11-19
> > > **Author Response to Reviewer vMKq**
> > >
> > > We thank the reviewer for acknowledgement of our response, and the continued engagement with our work.
> > >
> > > >This seems potentially somewhat unfair, as it results in an option that is always not correct. Perhaps the users would have leaned toward the correct answer, but were unsure, and instead selected the "unclear" option instead.
> > >
> > > The first (and foundational) question we wished to ask in our paper was "does outputting this visual mask intermediate task representation actually help humans disentangle "what" (behaviour-invariant) vs. "how" (behaviour-variant) features? In order to ensure that we were accurately capturing human certainty at either selecting the correct (or incorrect) response, we felt that including "unclear" would allow us to capture this uncertainty (which itself provides a gradient signal since if one condition produced certain results while the other produced uncertain, that is still a indication of the relative usefulness of one method vis-a-vis the other). Both the reviewer's proposed survey design method and our chosen method are valid design choices used within HCI/HAI human subject experiments.
> > >
> > > >...even if we can see a particular failure, I have concerns that it remains unclear how to augment the data in order to address this failure in general.
> > >
> > > Our main goal with this work is to instantiate one type of useful error-diagnostic framework that we can reasonably deploy given knowledge of model failure -- in visual navigation domains, this would distill down to object-centric visual features that would induce changed behaviours i.e. actions ("how" error) vs. those that would not i.e. features that would only result in scene, and not action, changes ("what" error). Generative models, as we know them to be today [1] are capable only of modifying scenes. For example, in the referenced paper, we see the ability to generate changed visual features based on image-specific feature specifications (e.g. a gray chair -> a red chair). This is **precisely** the type of feature-level augmentation our method requires, and the sufficient type of augmentation strategy that would result in the more efficient finetuning we show. What is key is that despite the success of these models at modifying scenes, *we would still require human users to perform demonstrations for feature changes that required novel actions*, such as the red chair now in a new location. Hence, assuming only the documented capabilities of generative models today, we can leverage our method to perform *scene augmentation* given knowledge of behaviour-invariant features (without a human) rather than requiring the full sampling of demonstrations (scenes + actions) that would only a human demonstrator would be able to provide. In other words, generative models performing visual-feature augmentation can result in data-efficient finetuning **if and only if** we are able to correctly identify that the shifted feature is *behaviour-invariant*, for otherwise we need novel actions irrespective. Our method provides a way to reduce human effort for generating unnecessary actions when scene-level modifications are sufficient.
> > >
> > > *References
> > > [1] Kawar, Bahjat, et al. "Imagic: Text-based real image editing with diffusion models." arXiv preprint arXiv:2210.09276 (2022).

---

> > > > ### Comment · Reviewer_vMKq · 2022-12-08
> > > > **Response**
> > > >
> > > > I really appreciate the further clarification, but still have concerns about the generality of the approach. I understand that the proposed approach could indeed reduce human feedback even in 3D settings, if we assume strong generative models that can accurately modify a scene without introducing artifacts that confuse an RL policy. However, I am still concerned that this sort of failure mode is somewhat specific and cannot be widely applied. In other failure modes, the agent may simply fail to move in the right direction or may wander around, which this framework does not particularly have any solutions for.

---

### Official Review · Reviewer_sTo9 · 2022-10-30

**Confidence:** 4
**Correctness:** 4
**Technical Novelty And Significance:** 3
**Empirical Novelty And Significance:** 3
**Recommendation:** 6

**Clarity, Quality, Novelty And Reproducibility:**

Clarity:

- The paper is well written.

Quality:

- The paper tackles an interesting problem.

Novelty:

- The reviewer is not aware of the similar problem setting discussed in the literature.



**Strength And Weaknesses:**

Strength:

- The paper is well written.
- The paper tackles an interesting problem i.e., how we can leverage the agent's internal representations to "do" interventions at the task level such that after training the agent's on the modified task distribution, the agent adapts efficiently.

Weakness:

- The main weakness is it may not be always feasible to construct "errors" based on internal representations on more complicated tasks (still the idea of parametrically augmenting the task distribution is interesting).
- The paper assumes access to the generative model capable of manipulating the factors of variation (and be able to construct new tasks). This limitation is already discussed in the paper, and the reviewer is hopeful that the future work should be able to leverage advances in generative modelling.

**Summary Of The Paper:**

The paper tackles an interesting problem. The basic idea is to use "privileged" feedback by the humans in the loop to improve adaptation of RL agents. The paper imposes a "constraint" on the agent's representations which helps the human user  to diagnose the agent’s failure mode. The paper tackles two kinds of "failure model": how error and what error. Based on the feedback, the human "parametrically" augments data from tasks (and thus the paper assumes access to the ground truth generative factors of variation) which may facilate adaptation and generalisation.

COMMENTS AFTER REBUTTAL:

The reviewer has read the reviews by other reviewer, as well the author's feedback. I retain my rating/recommendation.

**Summary Of The Review:**

The paper tackles an interesting problem: being able to diagnose why the agents are facing and based on the "surgery" being able to construct new tasks such that the resulting agents can adapt/transfer efficiently.

---

> ### Author Response · Authors · 2022-11-13
> **Author Response to Reviewer sTo9**
>
> We thank the reviewer for the time spent in providing feedback, and the conditional score of acceptance. Please refer to the general response regarding the reasonableness of assuming a generative model for scene generation.
>
> >**Q1**: The main weakness is it may not be always feasible to construct "errors" based on internal representations on more complicated tasks (still the idea of parametrically augmenting the task distribution is interesting).
>
> **A1**: We thank the reviewer for acknowledging the novelty of parametrically augmenting the task distribution based on human feedback, and wholly agree that finding additional methods for “diagnosing errors” in different domains is an important area of study. In our paper, we wished to instantiate one full interactive loop of our framework, including 1) demonstrating the ability of a common interpretability method to elucidate usable feedback, 2) scoping of a visual navigation domain such that the behaviour-invariant vs. dependent parameters were able to be highlighted by the method, 3) running a full human-diagnostic and feedback experiment to ensure this concept worked on real users, and 4) empirically demonstrating how this information can be leveraged to improve model performance. We believe that there is room to innovate on each of these axes, one of which is to formally conduct a comparative study on which types of feature disentanglement methods are best suited for constructing the specific “errors” required for each domain faced (e.g. object type may be a behaviour-invariant parameter for visual navigation, which is the task domain that we study in our paper, but would certainly be behaviour-dependent for domains involving object manipulation).

---

### Official Review · Reviewer_hcEg · 2022-10-31

**Confidence:** 4
**Correctness:** 3
**Technical Novelty And Significance:** 2
**Empirical Novelty And Significance:** 2
**Recommendation:** 5

**Clarity, Quality, Novelty And Reproducibility:**

**Clarity**
- As mentioned before, the paper is a bit hard to parse. Specifically, the ordering in Section 5 is a bit weird, 5.1 and 5.2 introduce tasks but then 5.3 and 5.4 discusses results. There is no parallel clean structure separating setups from results.
- Moreover, it seems the framework in Section 4 is discussed for specific tasks, and these tasks are introduced later in Section 5. I would recommend focusing on these two tasks from the outset if possible
- Section 3.2 is unclear including its section title "EFFICIENT ADAPTATION OF DISTRIBUTION SHIFT". It seems that an overview to the solution to the distribution shift problem is discussed in the second paragraph.
- In Section 4, the authors haven't formally defined on what they mean by "intermediate task representations".

**Details Of Ethics Concerns:**

The study involves research with human subjects in the form of their feedback. While it seems that the paper adequately addresses ethical issues, it might be good to have at least one review from an expert.

**Strength And Weaknesses:**

**Strengths**
- The paper considers a very important problem of incorporating human feedback to improve machine learning systems, in particular, to improve the performance of imitation learning policies.
- Experimental results are interesting when compared with a naive baseline. However, the paper misses a crucial baseline (see 2nd bullet in weaknesses).

**Weaknesses**
- The paper is a bit hard to parse. Mainly the organization in Sections 4 and 5 is a bit confusing. Please refer to the detailed comments below.
- In both the settings described, since authors are mainly choosing from two kinds of augmentations (depending on what and how failure), what if we could train a model with both the augmentations in the first place (without asking for human feedback)? Did authors try this in their experiments?
- Do authors use attention to get feedback from humans? While in many tasks attention masks do seem to correlate with models' final output, in very general cases, using attention to get an explanation for the model prediction can be misleading. Refer to [1,2,3]. I encourage authors to add a discussion and contrast their use case with studies in these references.
- It is unclear how the proposed framework may generalize beyond the two tasks considered in the paper where the disentanglement between how and what is not as easy. A more careful discussion of what kinds of problems the current framework can tackle will strengthen the paper.


[1] Pruthi et al. Learning to Deceive with Attention-Based Explanations. ACL 2020.

[2] Jain and Wallace. Attention is not Explanation. NAACLE 2019.

[3] Wiegreffe and Pinter. Attention is not not Explanation. EMNLP 2019.

**Summary Of The Paper:**

The paper tackles the problem of multi-task policies failing in the presence of distribution shifts. Authors disambiguate between goal and policy failures as the agent failing because it can’t correctly infer the desired goal or because it doesn’t know how to take an action toward achieving the goal. This work incorporates human feedback to diagnose the what versus how failure for adapting policies under distribution shift.

This work proposes a framework to incorporate human feedback by leveraging agents’ intermediate task representation.  Experiments with human users across discrete and continuous control tasks show that (i) intermediate task representations help human users in identifying failure mode; and (ii) subsequent empirical performance gains can be obtained in those tasks by leveraging human feedback.

**Summary Of The Review:**

Overall, while the paper tackles an important problem, the simple baseline of using all the augmentations is missing in the comparison. Moreover, using an attention mask can be unreliable in general and authors should refer to the added references above.

Since the proposed framework is also not general (at least right now it is not clear how to instantiate it under different tasks than the two considered in the work), I am leaning towards rejection. I would be happy to change my score if I share any misunderstanding that authors could clarify during rebuttals.

---

> ### Author Response · Authors · 2022-11-13
> **Author Response to Reviewer hcEg**
>
> We thank the reviewer for the time spent in providing feedback. Please refer to the general response for answers regarding generalizability of the framework to additional tasks, as well as current and proposed revisions for discussion on visual masks for interpretability and writing clarity in Sections 4 and 5.
>
> >**Q1**: In both the settings described, since authors are mainly choosing from two kinds of augmentations (depending on what and how failure), what if we could train a model with both the augmentations in the first place (without asking for human feedback)? Did authors try this in their experiments?
>
> **A1**: This is an excellent question and one that we cover in the first part of our general response.
>
> >**Q2**: Do authors use attention to get feedback from humans? While in many tasks attention masks do seem to correlate with models' final output, in very general cases, using attention to get an explanation for the model prediction can be misleading. Refer to [1,2,3]. I encourage authors to add a discussion and contrast their use case with studies in these references.
>
> **A2**: This is also an excellent question and one that we felt is a primary motivator for the specific method instantiation presented in our work: as the reviewer identified, the widespread use (and questionable validity) of attention masks as an explanatory tool is a question that is well studied in the interpretability community. However, we instead are interested in the question: “is attention, even if incorrect, useful for soliciting USABLE feedback from a human by exhibiting identifiable errors under specific parametric shifts for improving models?” In fact, the very ability of our method to disentangle errors of “what” vs. “how” is contingent upon the outputted attention mask showing more accurate “explanations” in some error scenarios (how) vs. others (what), for this inconsistency is the “diagnostic tool” that allows a human user to more accurately identify the behaviour-invariant tasks from the behaviour-dependent ones. We regret that this point was not made more clear in our paper, and propose this in our writing clarifications above.
>
> >I would be happy to change my score if I share any misunderstanding that authors could clarify during rebuttals.
>
> We hope that we were able to adequately revise misclarifications for both points and have addressed both of the concerns highlighted above!

---

> > ### Comment · Reviewer_hcEg · 2022-11-19
> > **Response**
> >
> > I thank the authors for their effort during the rebuttal and for providing clarifications. In particular, the authors claim that
> >
> > > our generative model for modifying states is not enough, since we require both new states that may vary largely from existing states as well as new behaviours (we do not assume the ability to generate entirely new actions). ?
> >
> > I completely agree with this but then it would be good to show a practical application where they can transfer this intuition. Overall, it is still unclear how the proposed framework may generalize beyond the two tasks considered in the paper where the disentanglement between how and what is not as easy.
> >
> > It is unclear if the authors have made the changes proposed in the "Second revision".
> >
> > Minor suggestion: It would have been easier for the reviewers to go through the updates if the changes in the main paper were highlighted with a different color.

---

### Author Response · Authors · 2022-11-13
**General Response (1/3)**

We would like to thank all reviewers for time spent in giving excellent comments. We felt that reviews were fair, constructive, and echoed shared sentiments.

**Summary of positive feedback**

All reviewers agree that we have identified an important and practical problem to the sequential learning community: how to diagnose policy failure modes when faced with parametric distribution shifts and incorporate human feedback to ultimately improve the system. We are heartened to hear that reviewers found our idea of leveraging human users to diagnose and parametrically augment task data “new” and “indeed original”.

Reviewers raised concerns regarding the generalizability of our proposed framework to more complex domains where 1) it may be difficult to parametrically generate desired task data and 2) “what” and “how” errors may be more difficult to disentangle. We first begin by clarifying the details and assumptions made in our method for data augmentation, and then address the above two concerns in turn.

**“How exactly is data generated for fixing “what” vs. “how” errors?” [vMKq]. “Why can’t we train a model with this information without human feedback?” [hcEg].**

We thank the reviewers for pointing out unclear details relating to our generative model, and have provided clarifications in a revised version of the paper in Sections 3.2 and 4.3 (uploaded).

In Section 3.2, which we have retitled “Adaptation for Behaviour-Invariant vs. Dependent Tasks”, we clarify the difference between shifted parameters that only require object-centric modifications to states such as navigating to a new goal object or color (i.e. behaviour-invariant or “what” tasks) vs. parameters that require new data capturing (possibly large) changes in both states and actions such as navigating to a new location (i.e. behaviour-dependent or “how” tasks). *We only assume a generative model capable of modifying the former.* In the “what” case, we leverage existing demonstrations to perform “surgical” object manipulation of the new goal into our existing states (an assumption that is reasonable in object-centric domains like visual navigation or goal-reaching, where behaviour for navigating a large environment towards different objects is not unique [1-3]) and re-use their corresponding actions. This adaptation strategy requires 0 novel demonstrations from the human user, since we are entirely leveraging existing data via modifications of our generative model to existing states. In the “how” case, our generative model for modifying states is not enough, since we require both new states that may vary largely from existing states as well as new behaviours (we do not assume the ability to generate entirely new actions). We therefore consider collecting “novel demonstrations” as a form of human feedback (which we simulate through an expert demonstrator), but then retain the ability to leverage these new states and actions to augment existing demonstrations with. Therefore, we assume the ability to augment “what” or behaviour-invariant tasks via our generative model, but not “how” or behaviour-dependent ones, setting up the problem to be one of *minimizing the number of human demonstrations required for finetuning a particular distribution shift*. These details are also clarified in Section 4.3.

In Section 5.4 (p. 8), we detail our baselines and their corresponding augmentation strategies. The “No Human” baseline details the case where we have identified only that an error has occurred on the test task, but receive nothing from a human (i.e. we don’t know the parameter of shift nor have the ability to query for new demonstrations). Thus, under our assumptions of the reasonableness of a generative model’s ability to modify existing object-centric states but not generate new actions, the reviewer’s suggestion of a baseline with half “what” and “how” demonstrations requiring no human intervention is not possible, for the very act of creating new demonstrations requires human feedback of new actions outside the capability of our generative model. Our “Naive H” and “Informed H” baselines detail the cases when new demonstrations covering new actions are able to be provided for finetuning.

---

> ### Author Response · Authors · 2022-11-13
> **General Response (2/3)**
>
> **Common concern 1: generative model assumption**
>
> Reviewers expressed a valid concern regarding the reasonableness of assuming a generative model for modifying the task data for augmentation, particularly in more complex real-world settings beyond the 2D tasks studied in the paper. On this point, the authors hope the above clarifications regarding our assumption of object-centric state modifications are helpful and we, along with reviewer sTo9, remain optimistic by the many recent advancements on improved capabilities of generative models for scene manipulation.
>
> The field of generative modeling is exploding before our eyes: the ability to selectively replace specific objects or features in realistic scenes has elicited very impressive results. It appears to be not a question of if, but when, such methods will be leveraged towards “surgical” task-conditioned state generation of tasks for robot learning. However, the applicability of such methods to solving specific learning problems faced by failed policies is limited by our ability to identify the desired parameters in our test environment that we need to fix (i.e. we may be able to generate a scene, but how do we know that this data pinpoints the underlying model failure? In other words, *for which problems can a generative model help us create data for vs. we are required to collect?*). We believe this is a challenge that human users who we deploy systems for are uniquely positioned to diagnose and help solve, i.e. *humans are really good at knowing which tasks require fundamentally new behaviours vs. not*. Therefore, we believe that the timing of our framework is quite auspicious and well-positioned to contribute to questions asked by the community on how to best leverage these models for downstream human-aligned learning. It is our hope that the generative model component of our framework provides very clear and exciting follow-up directions for those wishing to leverage advances in future model advancements for human-centric adaptation.
>
>
> **Common concern 2: use of an attention mask to disentangle errors in more complex domains**
>
> All reviewers raised valid concerns regarding the generalizability of attention masks to more complex domains. These included comments about “the framework’s ability to generalize beyond the two tasks considered in the paper” [hcEG], the feasibility of “constructing “errors” based on internal representations on more complicated tasks” [sTo9], and uncertainty regarding “how to add a visualizable or human-interpretable intermediate representation for other general tasks” [i8yl]. We acknowledge that these are excellent concerns and that simple visual masks as the sole error diagnosis tool will likely not suffice in all complicated robotic settings. As we wrote in Limitations on p.9, there will always be “parameters that are difficult to practically disentangle through visual attention alone such as object occlusion, shape and texture, and partially observed scenes”. However, we remain hopeful of our framework’s applicability to real-world problems of value for the following two reasons:
>
> First, although visual attention is indeed a limited error diagnosis tool for disentangling many complex features, it remains a useful tool for identifying behaviour-invariant features in visual navigation tasks such as object type and color (e.g. a home robot identifying and navigating to the fridge in a room or a robot gripper needing to identify and reach towards a particular object on a table). Although the tested domains in our paper live within 2D Gridworlds, our method’s ability to operate on any task defined by object-centric RGB images (even relatively sparse tasks composed of sequences of sub-tasks like the Minigrid experiments) appears to hold promise as a widely-applicable diagnosis tool incorporated within more complex robotic navigation tasks, where object locations in 3D space can be reasonably identified by spatial attention. *We believe that these existing applications of visual masks to complex real-world settings are well documented [1-3] and highlight the potential for our method’s use as a more widely applicable error diagnosis tool for disentangling errors in object-centric navigation tasks.* However, it is certainly true that for visual parameters in tasks that are behaviour-dependent, such as end-effector manipulation of new object shapes, we would likely need alternate error diagnosis tools. We believe a comparative study of which visual disentanglement methods are best suited for different domains to be a question most appropriate for future work, and suggest a formal study of new methods via our framework. We recognize that the scoping of which tasks to apply our method to (identifying behaviour-invariant parameters for object-centric visual navigation or goal reaching) was not defined well in the paper, and will rewrite our introduction and problem framing accordingly.

---

> > ### Author Response · Authors · 2022-11-13
> > **General Response (3/3)**
> >
> > Second, while we instantiate visual attention as one particular solution for the error diagnosis component of our method, our belief is that our paper can serve as *a general framework for incorporating many other error diagnosis tools for feature-disentanglement*. For example, the field of interpretability has long explored mechanisms for generating intermediate model representations to humans, and newer work holds promise for integration with deep learning methods such as outputting language descriptions of complex visual scenes [6] and graph-based explanations [4,5]. Yet, within such a rich landscape of many different modality-specific tools, there remains little exploration on how to leverage such tools to close the loop in improving the system for an end-to-end task (our paper to the best of our knowledge provides *one of the first full interactive pipelines verified with real human feedback* that leverages a widely used interpretability mechanism for human error diagnosis and model improvement within a particular task setup). It is our hope that this work offers a path for the learning community towards an exciting and promising direction of future study in how to think about incorporating interpretability methods as a human-in-the-loop parametric error diagnosis and data improvement tool. In this way, we are offering an alternative use case for thinking about the impact of interpretability methods: not just as a tool to reveal a “ground truth” explanation of the model, but rather as a “window”, even if incorrect, into the model’s internal representation that is useful for diagnosing and fixing the failure point.
> >
> > **Framework contribution**
> >
> > Ultimately, we believe that a key contribution of our paper is introducing a novel framework for thinking about *how to leverage human feedback in an efficient manner to help improve systems*, a paradigm that we believe is critical to the future development of scalable, efficient human-aligned systems. In this paper, we demonstrated a proof-of-concept of this interactive pipeline by instantiating one full loop of this framework in a visual navigation domain beginning with environment and data generation, incorporation of a common interpretability tool for communicating failure modes to the human, a real user study testing the human’s ability to behaviour-invariant from dependent parameters (even with an incorrect mask), and the ability to leverage that feedback towards ultimate improvement of the system. We are hopeful that the individual components of our framework (error diagnosis tool and better generative modeling) can all easily be replaced by future work in both lines of inquiry that elicit better feature disentanglement of more complex tasks. We will rewrite our introduction and discussion to reflect such emphasis.
> >
> > **First Revision (Uploaded)**
> >
> > In light of the writing clarity questions asked by reviewers, we have uploaded a first revision of our paper, mostly clarifying experimental details. These include (in order of appearance):
> > 1. Clarification of Equations 1 and 2 per Reviewer vMKq’s feedback.
> > 2. Re-writing of Section 3.2 as “Adaptation for Behaviour-Invariant vs. Dependent Tasks”, clarifying the relationship between the generative model, train and test distributions, and human feedback. We also emphasize our assumptions.
> > 3. Clarification of how “what” and “how” augmentations are generated (Section 4.3).
> > 4. Clarification of the correctness calculation for human experiments (Section 5.3).
> >
> > **Second Revision (Proposed)**
> >
> > Going forward, we propose an additional revision highlighting larger structural changes (in order):
> >
> > 1. Reworking of our introduction to emphasize our framework contribution for leveraging humans to diagnose parametric failures, with a clear scoping of the proposed method as an instantiation for disentangling “what” and “how” errors in visual navigation or goal-reaching tasks.
> > 2. Including a brief discussion in the related work on how visual masks may or may not be accurate in providing “true explanations” of the model, yet can be leveraged by the human to infer information useful for identifying behaviour-invariant vs. dependent parameters.
> > 3. Reworking of Section 5 to separate our method in 5.1 and 5.2 from results in 5.3 and 5.4.
> > 4. Adding the need for more complex error diagnosis tools and experimental validation in limitations.
> > 5. Expand the discussion on generative modeling to suggest specific ideas for incorporation (e.g. generating new objects within common household robot navigation tasks) [10].

---

> > > ### Author Response · Authors · 2022-11-13
> > > **References**
> > >
> > > **References**
> > >
> > > 1. Seymour, Zachary, et al. "Maast: Map attention with semantic transformers for efficient visual navigation." 2021 IEEE International Conference on Robotics and Automation (ICRA). IEEE, 2021.
> > > 2. Mayo, Bar, Tamir Hazan, and Ayellet Tal. "Visual navigation with spatial attention." Proceedings of the IEEE/CVF Conference on Computer Vision and Pattern Recognition. 2021.
> > > 3. Chen, Shi, and Qi Zhao. "Attention to Action: Leveraging Attention for Object Navigation." The British Machine Vision Conference. 2021.
> > > 4. Daruna, Angel, Devleena Das, and Sonia Chernova. "Explainable Knowledge Graph Embedding: Inference Reconciliation for Knowledge Inferences Supporting Robot Actions." arXiv preprint arXiv:2205.01836 (2022).
> > > 5. Bewley, Tom, and Jonathan Lawry. "Tripletree: A versatile interpretable representation of black box agents and their environments." Proceedings of the AAAI Conference on Artificial Intelligence. Vol. 35. No. 13. 2021.
> > > 6. Hernandez, Evan, et al. "Natural Language Descriptions of Deep Visual Features." International Conference on Learning Representations. 2021.
> > > 7. Lipman, Yaron, et al. "Flow Matching for Generative Modeling." arXiv preprint arXiv:2210.02747 (2022).
> > > 8. Berner, Julius, Lorenz Richter, and Karen Ullrich. "An optimal control perspective on diffusion-based generative modeling." arXiv preprint arXiv:2211.01364 (2022).
> > > 9. Bautista, Miguel Angel, et al. "GAUDI: A Neural Architect for Immersive 3D Scene Generation." arXiv preprint arXiv:2207.13751 (2022).
> > > 10. Li, Chengshu, et al. "igibson 2.0: Object-centric simulation for robot learning of everyday household tasks." arXiv preprint arXiv:2108.03272 (2021).

---

### Public Comment · ~Lin_Guan1 · 2023-02-11
**A highly relevant line of work**

This interesting paper discusses the potential use of visual attention maps in facilitating the diagnosis of failure modes and allowing more informative feedback from humans. A relevant line of work has explored the use of human saliency maps (e.g., hand-annotated task-relevant regions or human gaze) to provide stronger human supervision. This idea has been applied to human-in-the-loop RL [1], IL [2], and other ML tasks [3, 4]. An interesting question here is whether it is more effective to have humans correct & regularize the model's attention directly, or to do it indirectly (e.g., as in this work, by indicating the failure mode). It may be helpful to include some discussion or empirical comparisons regarding this in the paper.


Best Regards,
Lin



[1] Guan, Lin, Mudit Verma, Suna Sihang Guo, Ruohan Zhang, and Subbarao Kambhampati. "Widening the pipeline in human-guided reinforcement learning with explanation and context-aware data augmentation." NeurIPS 2021.

[2] Zhang, Ruohan, Zhuode Liu, Luxin Zhang, Jake A. Whritner, Karl S. Muller, Mary M. Hayhoe, and Dana H. Ballard. "Agil: Learning attention from human for visuomotor tasks." ECCV. 2018.

[3] Ross, Andrew Slavin, Michael C. Hughes, and Finale Doshi-Velez. "Right for the right reasons: Training differentiable models by constraining their explanations." IJCAI 2017.

[4] Rieger, Laura, Chandan Singh, William Murdoch, and Bin Yu. "Interpretations are useful: penalizing explanations to align neural networks with prior knowledge." ICML, 2020.

---

### Decision · Program_Chairs · 2023-01-20

**Decision:**

Reject

**Justification For Why Not Higher Score:**

The paper is based on several strong assumptions, and the experiments are performed using simplistic environments and tasks. So, acceptance is not recommended.

**Justification For Why Not Lower Score:**

N/A

**Metareview: Summary, Strengths And Weaknesses:**

The idea of the paper is to disentangle different failure modes of an agent. More specifically, the paper proposes to distinguish between “what” (i.e. what the goal is ) and “how” (i.e. how to achieve the goal) failures. The proposed method uses attention to provide a human interpretable representation for the internal state of the agent. Leveraging the feedback from humans leads to performance improvements.

The main strength of the paper is that it addresses an important and interesting problem. Soliciting feedback from people and inferring how to use the feedback has a lot of potential applications in real world scenarios.

The main weakness of the paper is that it is based on strong assumptions. For example, it assumes there is a generative model that generates scenes. This assumption is quite strong and it probably works only for very simple scenarios studied in this paper (2D grid world). Another assumption is that the attention model can be used to communicate the state of the agent. Additionally, the paper assumes that the “how” and “what” errors can be disentangled easily.

The reviewers participated in a discussion. The reviewers believe it is not clear how disentanglement between “how” and “what” is possible beyond the simplistic tasks considered in this paper. Regarding the generative model assumption, the authors argue that the generative models are becoming stronger so they can be used. While this statement is true, the reviewers want to see how they work in action. The current generative models are still far from the requirements of the proposed method. Finally, intermediate representations do not necessarily preserve any semantic meaning when large models are used. So, using attention is not necessarily useful for other architectures.

Overall the reviewers did not find the authors’ responses convincing. The AC also read the reviews and the rebuttal carefully and agrees with the reviewers. Therefore, rejection is recommended.